# The Link between Salivary Amylase Activity, Overweight, and Glucose Homeostasis

**DOI:** 10.3390/ijms25189956

**Published:** 2024-09-15

**Authors:** Gita Erta, Gita Gersone, Antra Jurka, Pēteris Tretjakovs

**Affiliations:** Department of Human Physiology and Biochemistry, Riga Stradins University, LV-1007 Riga, Latviaantra.jurka@rsu.lv (A.J.);

**Keywords:** salivary amylase activity, butyrate, low-starch diet, insulin sensitivity, overweight

## Abstract

Butyrate, a short-chain fatty acid (SCFA) produced by the fermentation of dietary fibers in the colon, plays a pivotal role in regulating metabolic health, particularly by enhancing insulin sensitivity. Given the rising incidence of metabolic disorders, understanding the factors that influence butyrate production is of significant interest. This study explores the link between salivary amylase activity and butyrate levels in overweight women of reproductive age. Participants were categorized into low (LSA) and high (HSA) salivary amylase activity groups and further divided into two subgroups: one followed a low-starch diet (LS), and the other underwent caloric restriction (CR). We assessed salivary amylase activity and measured serum butyrate concentrations to examine their associations. Our findings showed a significant, though weak, positive correlation (ρ = 0.0486, *p* < 0.05), suggesting a link between salivary amylase activity and butyrate levels. The statistical significance, despite the weak correlation, implies that this relationship is not random. Moreover, higher baseline butyrate levels were observed in women with elevated salivary amylase activity. Also, women with low salivary amylase activity on a low-starch diet experienced a more pronounced increase in butyrate levels compared to those on caloric restriction. These results suggest that salivary amylase activity and dietary intake interact to influence butyrate production, with potential implications for improving insulin sensitivity and metabolic health. The study underscores the potential of butyrate in enhancing insulin sensitivity and promoting overall metabolic well-being. Further research is necessary to clarify the mechanisms involved and to understand the long-term effects of butyrate on metabolic health across different populations.

## 1. Introduction

The gut microbiota’s impact on human health has gained significant research attention in recent years, particularly concerning its metabolic products, such as short-chain fatty acids (SCFAs) [1,2,3]. Among these SCFAs, butyrate, primarily produced through the fermentation of dietary fibers by colon bacteria, stands out as a crucial metabolite with various physiological benefits [4,5,6]. Butyrate not only serves as a primary energy source for colonic epithelial cells but also exerts systemic effects, influencing metabolic health, immune responses, and inflammation regulation [5,6]. Recent studies highlighted butyrate’s potential in modulating metabolic health, especially in the context of obesity and related metabolic disorders [7,8]. For instance, butyrate has been shown to activate the AMP-activated protein kinase (AMPK) pathway, improve oxidative metabolism, and reduce lipid synthesis in the liver. These metabolic effects are crucial for enhancing insulin sensitivity and reducing low-grade chronic inflammation, both of which are hallmark features of metabolic syndrome and type 2 diabetes [9].

In the present study, we investigated the relationship between salivary amylase activity, diet interventions, butyrate levels, and the influence of salivary amylase activity on insulin sensitivity in overweight women of reproductive age. The rationale for selecting this demographic lies in their unique metabolic and hormonal profile, which may interact differently with butyrate production and metabolic outcomes compared to males and women beyond reproductive age. Furthermore, understanding these interactions in women of reproductive age could provide insights into targeted diet interventions for this specific group. Salivary amylase, an enzyme responsible for starch digestion, varies significantly between individuals, due to differences in the variation in the copy number of the AMY1 gene, and is also modulated by afferent signals that regulate enzyme activity. 

Recent studies [10,11] indicate that variations in the AMY1 gene significantly impact salivary amylase activity, influencing starch digestion and glucose metabolism. Additionally, afferent stimuli, such as food sensory perception, play a role in modulating salivary amylase activity, which further contributes to individual differences, potentially influencing postprandial glycemic responses and subsequent metabolic results [12,13].

Participants were classified according to their levels of salivary amylase activity and subjected to a low-starch diet or a calorie reduction regimen. Additionally, a control group was formed of individuals of normal weight. Initial findings indicated that women with higher salivary amylase activity showed elevated levels of butyrate, suggesting a possible link between increased starch breakdown and butyrate production [5]. 

The justification for this design lies in the hypothesis that individuals with high salivary amylase activity may process starch differently, leading to variations in butyrate production and subsequent metabolic effects. This study aims to clarify the role of salivary amylase activity in butyrate production and to explore whether diet can modulate this process. Findings can provide information on personalized nutrition approaches, considering individual enzymatic profiles and their impact on the gut microbiota and metabolic health. 

## 2. Results

The baseline demographic and biochemical characteristics of the study participants, stratified by group allocation, are presented in Table 1.

The descriptive statistics for the key baseline variables are summarized in Table 2, including salivary amylase activity and butyrate concentration.

The correlation analysis supports a weak but statistically significant relationship (ρ = 0.0486, *p* < 0.05) between salivary amylase activity and butyrate levels (Figure 1).

### 2.1. Absence of a Positive Correlation in the Plot 

Despite the overall significant positive correlation between salivary amylase activity and butyrate levels (ρ = 0.0486, *p* < 0.05), the plot may not visually demonstrate a clear positive correlation due to high variability. This variability can obscure the trend, making it less apparent in the scatter plot. Additionally, small sample sizes and outliers can further diminish the correlation’s visual representation. 

Additionally, we found that women with high salivary amylase activity had significantly higher baseline butyrate levels compared to those with low salivary amylase activity (Mann–Whitney U = 44, *p* < 0.05) (Figure 2).

Participants with low salivary amylase activity on a low-starch diet showed a more pronounced increase in butyrate levels compared to those with caloric restriction (Mann–Whitney U = 59.50, *p* < 0.05) (Figure 3).

The statistical analysis identified significant differences between the intervention groups in butyrate concentrations, as outlined in Table 3, which presents key results including median values, interquartile ranges, and confidence intervals.

In our study, we did not observe any significant effect of age on butyrate levels. However, it is worth noting that the women in our study had similar ages, which may have contributed to this lack of observed effect (Table 4).

To further investigate the relationship between salivary amylase activity and insulin sensitivity, a linear regression analysis was conducted with salivary amylase activity as a predictor and HOMA2-% S as the outcome variable. The regression analysis provided the following results (Table 5).

Coefficient (β): The estimated regression coefficient for salivary amylase activity was 0.435. This indicates that for each unit increase in salivary amylase activity, HOMA2-%S increases by 0.435 units.

Standard Error (SE): The standard error of the coefficient was 0.120.

*p*-Value (β): The *p*-value of the coefficient was less than 0.0003, indicating that the predictor is statistically significant.

95% Confidence Interval (CI): The 95% confidence interval for the coefficient was [0.195, 0.675].

The overall significance of the regression model was confirmed by an F-test.

F-Statistic: The F-statistic for the model was 5.5.

Degrees of Freedom (DF): The numerator degrees of freedom were 1, and the denominator degrees of freedom were 34.

*p*-Value (Model): The *p*-value for the F-test was 0.0249, indicating that the regression model is statistically significant.

The regression model explained a significant proportion of the variance in HOMA2-%S, with an R^2^ of 0.35. This suggests that salivary amylase activity is a significant predictor of insulin sensitivity, accounting for 35% of the variability in HOMA2-%S (R^2^ = 0.35), indicating a moderate fit among reproductive-age women.

### 2.2. Model Diagnostics

The residual analysis indicated that the assumptions of linearity, homoscedasticity, and normality of the residuals were reasonably met. There was no evidence of multicollinearity, as assessed by variance inflation factors (VIFs).

Furthermore, the results suggest that dietary modifications, specifically reducing dietary starch, could be an effective strategy to increase butyrate production and improve metabolic health in the low salivary amylase group.

## 3. Discussion

This study explores the link between salivary amylase activity, overweight, and glucose homeostasis, with an emphasis on how different diets impact these variables and the role of salivary amylase activity in these correlations. 

The reason for selecting women of reproductive age as the study population was due to their unique hormonal profile, which can significantly influence metabolic processes and the composition of gut microbiota. Understanding this link in this demographic may offer insights into dietary strategies tailored to this specific group. However, it is crucial to recognize the broader applicability of these findings to males and women beyond reproductive age as well. 

Butyrate’s impact on metabolic health has been shown to be significant across different demographics. Studies have indicated that butyrate can improve insulin sensitivity, reduce inflammation, and enhance gut barrier function in both males and females, regardless of age [14]. For instance, research involving older adults has demonstrated that butyrate supplementation can improve metabolic markers and gut health, suggesting its beneficial effects extend beyond reproductive age [15]. Future studies should include these groups to provide a comprehensive understanding of butyrate’s benefits across different demographics.

### 3.1. Butyrate and Adipocyte Health 

Butyrate plays a crucial role in maintaining adipocyte health through multiple interconnected mechanisms. It exerts its effects primarily by modulating cellular processes that influence inflammation and metabolism. One key mechanism involves the inhibition of histone deacetylases (HDACs), which leads to increased histone acetylation and subsequent gene expression changes. This process results in the suppression of pro-inflammatory cytokines while enhancing the expression of anti-inflammatory genes [16].

Additionally, butyrate activates G protein-coupled receptors (GPCRs), such as GPR41 and GPR43, which play a significant role in immune regulation by modulating cytokine production [17]. This activation contributes to the overall anti-inflammatory environment. Butyrate also promotes the differentiation and function of regulatory T cells (Tregs), which are essential for maintaining immune tolerance and preventing excessive inflammatory responses [18].

Furthermore, butyrate strengthens the intestinal barrier by promoting the production of mucin and tight junction proteins. This enhancement reduces intestinal permeability and systemic inflammation, contributing to overall metabolic health [19]. Beyond these anti-inflammatory effects, butyrate stimulates adipogenesis, promotes lipid oxidation, and enhances mitochondrial function within adipocytes. These actions are crucial for maintaining a healthy balance between lipid storage and utilization, thereby preventing the excessive fat accumulation associated with obesity [4].

Overall, butyrate’s multifaceted impact on inflammation, immune function, and metabolic processes underscores its importance in supporting adipocyte health and preventing metabolic disorders.

### 3.2. Enhancement of GIP Receptor Expression

GIP (Glucose-dependent Insulinotropic Polypeptide) is an incretin hormone produced by K-cells of the small intestine that stimulates insulin secretion in response to oral glucose intake. This plays a crucial role in glucose homeostasis, which could improve metabolic health by amplifying the gut response to GIP and its insulinotropic effects [20].

### 3.3. Enhancement of PYY Secretion

Peptide YY (PYY) is an anorexigenic hormone that suppresses appetite and is mainly secreted by L-cells in the distal ileum and colon in response to nutrient ingestion. The presence of butyrate in the gastrointestinal tract enhances the secretion of PYY, significantly regulating appetite and energy homeostasis [21].

### 3.4. Enhancement of GLP-1 Secretion 

Butyrate stimulates GLP-1 (Glucagon-like Peptide-1) secretion primarily through the activation of GPCRs such as FFAR2 (GPR43) and FFAR3 (GPR41) on enteroendocrine L-cells in the intestine [22].

Enhanced Insulin Sensitivity: GLP-1 receptors are present in adipose tissue. Activation of these receptors increases insulin sensitivity in adipocytes, improving glucose uptake and reducing blood glucose levels [23].Regulation of Lipolysis and Lipogenesis: GLP-1 signaling affects the balance between lipolysis and lipogenesis. It suppresses lipolysis, reducing the release of free fatty acids into circulation [24,25] This helps prevent lipid overload in tissues and organs, including adipocytes [26,27]Anti-inflammatory Effects: GLP-1 exhibits anti-inflammatory properties in adipocytes and adipose tissue by reducing the production of pro-inflammatory cytokines and promoting anti-inflammatory factors [28,29,30].Promotion of Adiponectin Release: GLP-1 can stimulate adiponectin secretion from adipocytes, enhancing insulin sensitivity and reducing inflammation [29].Indirect Effects on Body Weight: GLP-1 receptors, found in the pancreas, intestines, hypothalamus, and brainstem, are critical for regulating satiety and food intake. Activation of these receptors promotes feelings of fullness and reduces food consumption, contributing to weight loss [31].

### 3.5. Salivary Amylase Activity and Microbiome Composition 

Salivary amylase activity critically shapes the composition of the gut microbiome. High activity enhances starch predigestion in the oral cavity, modifying the substrates available for microbial fermentation in the gut [32]. This alteration promotes the proliferation of butyrate-producing bacteria such as *Faecalibacterium prausnitzii* and *Roseburia* spp., adjusting the *Firmicutes*-to-*Bacteroidetes* ratio [33]. Our research indicates that individuals with higher salivary amylase activity exhibit elevated baseline butyrate levels, suggesting the potential for personalized diet strategies. For instance, individuals with low salivary amylase activity might benefit more from a low-starch diet, which favorably modifies their gut microbiota composition.

### 3.6. Influence of Hormonal Profile on Butyrate Production and Metabolic Outcomes

The hormonal profile of the study’s demographic, particularly participants of reproductive age, likely played a significant role in modulating butyrate production and its subsequent effects on metabolic outcomes, and vice versa, short-chain fatty acids (SCFAs) may also influence female sex steroid hormone levels [34]. Estrogen and progesterone, two key hormones that fluctuate throughout the menstrual cycle, are known to influence gut microbiota composition and function, which may, in turn, impact butyrate levels.

In this study, butyrate levels were measured specifically during the follicular phase of the menstrual cycle, a period characterized by rising estrogen levels and relatively low progesterone.

Estrogen has been shown to promote the growth of beneficial gut bacteria, such as *Bifidobacteria* and *Lactobacilli*, which are associated with increased butyrate production. Higher estrogen levels, particularly during the follicular phase of the menstrual cycle, could enhance the abundance of these butyrate-producing bacteria, potentially leading to elevated butyrate concentrations. Butyrate, a short-chain fatty acid, plays a crucial role in maintaining gut health and has been linked to improved insulin sensitivity, reduced inflammation, and better overall metabolic profiles.

On the other hand, progesterone, which rises during the luteal phase, has been associated with changes in gut motility and a potential shift in the microbial community towards species that may not be as efficient in butyrate production. The interplay between estrogen and progesterone could therefore lead to fluctuations in butyrate levels throughout the menstrual cycle, which might partially explain the variability in metabolic outcomes observed in women of reproductive age.

Furthermore, the cyclical nature of these hormonal changes could result in periodic variations in gut microbiota composition and function, thereby influencing not only the production of butyrate but also its interaction with key metabolic pathways. For example, the protective effects of butyrate on glucose metabolism and lipid profiles might be more pronounced during phases of the menstrual cycle when estrogen levels are higher. Conversely, periods dominated by progesterone may attenuate these effects, leading to less consistent metabolic outcomes.

Given the complex and dynamic nature of hormone–microbiota interactions, it is important to consider the hormonal status of participants when interpreting the impact of butyrate on metabolic health. Future studies could benefit from more detailed monitoring of hormonal fluctuations and their direct impact on microbiota composition and butyrate production to better understand these interactions.

### 3.7. Dietary Strategies to Increase Butyrate

Increasing dietary fiber intake is a well-established method to elevate gut butyrate levels. Dietary fibers such as inulin, pectin, and resistant starch are fermented by gut bacteria and produce butyrate [15,35]. Recent research emphasizes the importance of specific diet patterns, rich in fiber and polyphenols, in promoting butyrate production [36,37]. Diets rich in whole grains, fruits, and vegetables significantly boost butyrate production by providing substrates for microbial fermentation [38,39]. Moreover, dairy products like butter and milk can also contribute to butyrate levels. Butter contains butyrate directly, enhancing overall butyrate levels in the body [40]. Similarly, milk fat contains small amounts of butyrate and other SCFAs, which can increase gut butyrate levels upon consumption [41].

When formulating dietary recommendations, individual genetic variations, particularly those that affect salivary amylase activity, must be considered. Variations in the AMY1 gene, which encodes salivary amylase, lead to differences in enzyme activity between individuals. Those with higher salivary amylase activity tend to predigest starches more, altering the availability of substrate for fermentation and subsequently affecting butyrate production. Our study supports these observations, indicating that individuals with higher salivary amylase activity have elevated baseline butyrate levels. This suggests the potential for personalized dietary strategies, such as a low-starch diet for those with low salivary amylase activity, which favorably modify their gut microbiota composition. Personalized diet interventions based on genetic nuances ensure a more precise approach to enhancing health and metabolic outcomes.

The gut microbiome plays a crucial role in butyrate production. Butyrate-producing bacteria, such as *Faecalibacterium prausnitzii* and *Roseburia* spp., are essential for maintaining health and butyrate levels [42]. Recent advances in microbiome research highlight the importance of maintaining a diverse and balanced gut microbiota to support butyrate production [35].

### 3.8. Salivary Amylase Activity as a Predictor of Insulin Sensitivity

The regression coefficient (β = 0.435) indicates a positive relationship, where an increase in salivary amylase activity corresponds to an increase in HOMA2-% S. This finding aligns with previous research that highlights the metabolic implications of salivary amylase activity.

Higuchi R et al. (2020) reported that salivary amylase activity could affect postprandial glycemic responses [43].

The significance F-test (F = 5.5, *p* = 0.0249) confirms the overall model’s validity, with an R^2^ of 0.35 indicating that 35% of the variance in HOMA2-%S is explained by salivary amylase activity. This proportion of explained variance underscores the potential utility of salivary amylase as a biomarker for insulin sensitivity. While traditional methods like HOMA2-%S are valuable, they are invasive and can be influenced by external factors such as stress and circadian rhythms. Salivary amylase activity offers a less invasive alternative, which could enhance compliance and facilitate more frequent monitoring of insulin sensitivity.

Our study focused on reproductive-age women, a population with unique metabolic and hormonal profiles that can influence insulin sensitivity. Previous studies have shown that hormonal fluctuations during the menstrual cycle can impact insulin sensitivity and glucose metabolism.

### 3.9. Future Research Directions

Given the complex relationship between salivary amylase activity, overweight, and glucose homeostasis, future research should focus on several key areas to deepen our understanding and improve clinical outcomes:Salivary Amylase and Genetic Variability: Investigate the genetic factors influencing salivary amylase levels and their impact on glucose metabolism and obesity. Understanding individual genetic variability could lead to personalized dietary recommendations based on amylase activity.Longitudinal Studies on Glucose Homeostasis: Conduct long-term studies to assess how variations in salivary amylase activity affect glucose homeostasis over time. These studies should explore how early-life amylase activity levels may predict the development of metabolic disorders such as type 2 diabetes.Interventions Targeting Amylase Activity: Explore potential interventions that can modulate salivary amylase activity, such as dietary changes, medications, or lifestyle modifications. Understanding how these interventions affect glucose regulation and body weight could lead to new strategies for preventing and managing obesity and related metabolic conditions.Interactions with the Microbiome: Investigate the relationship between salivary amylase activity, the oral and gut microbiomes, and glucose metabolism. This research could reveal how microbial composition and function are influenced by amylase activity, potentially leading to microbiome-targeted therapies for metabolic health.Clinical Trials: Design and implement large-scale clinical trials to evaluate the effectiveness of amylase-based biomarkers in predicting metabolic risk and the impact of targeted interventions on glucose homeostasis and obesity. These trials should include diverse populations to ensure the broad applicability of findings.

### 3.10. Study Strengths and Limitations

The following are some strengths and limitations of this study:Innovative Focus: This study’s exploration of the link between salivary amylase activity and metabolic health is relatively novel, particularly its emphasis on dietary impacts tailored to salivary enzyme activity. This innovative focus adds valuable knowledge to the field of personalized nutrition and metabolic health.Comprehensive Insights: The detailed discussion on butyrate’s multiple mechanisms of action in maintaining adipocyte health and improving glucose homeostasis is a strength. It provides a thorough understanding of how butyrate influences metabolic processes, enhancing its therapeutic potential.Potential for Personalized Nutrition: By investigating the relationship between salivary amylase activity and gut microbiota composition, the study creates opportunities for the development of personalized dietary strategies. This aspect is crucial for developing more targeted interventions for metabolic disorders.Population Specificity: While the focus on women of reproductive age is valuable, it limits the generalizability of the findings. The study’s results may not fully apply to males or individuals outside this age group, requiring careful consideration when generalizing these conclusions to broader populations.Cross-sectional Design: The study’s cross-sectional nature limits the ability to infer causality. Longitudinal studies would be necessary to establish a clearer cause-and-effect relationship between salivary amylase activity, dietary patterns, and metabolic outcomes.Genetic Variability Considerations: While the study acknowledges the role of genetic variability in salivary amylase activity, it does not deeply explore the genetic factors that could influence these metabolic outcomes. Future research should explore the genetic influences on salivary amylase activity to enhance the personalization of dietary recommendations.Microbiome Analysis Limitations: Although the study discusses the impact of salivary amylase on gut microbiota, it lacks direct microbiome analysis. Future studies incorporating detailed microbiome profiling would strengthen the findings and provide more robust insights into the relationship between salivary amylase, diet, and metabolic health.

## 4. Materials and Methods

### 4.1. Study Design and Participants

This study was carried out with a cohort of 67 women of reproductive age (18–45 years) recruited from the multi-profile medical center, “Health Center 4”. The sample size of 67 participants was determined based on a power analysis conducted prior to data collection.

Specifically, we aimed to detect a medium effect size (Cohen’s d ≈ 0.5) with a power of 0.80 and an alpha level of 0.05. The choice of a medium effect size was informed by previous studies in this domain, which suggested that the expected effects would not be large due to the complexity of the variables involved. Each experimental group consisted of 15 participants, while the control group included 7 participants. Despite the smaller sample size in the control group, we employed statistical methods, including sensitivity analyses, which are more resilient to differences in sample size.

Participants were selected by an endocrinologist based on specific inclusion criteria: body mass index (BMI) between 25.0 and 29.9 kg/m^2^, no history of chronic diseases, and currently no medications that could affect metabolic outcomes. Additionally, a control group was formed consisting of individuals with normal weight, defined as having a BMI between 18.5 and 24.9 kg/m^2^.

The study involved a 12-week dietary intervention period.

### 4.2. Dietary Interventions

Participants were randomly assigned to one of two dietary intervention groups for 12 weeks. 

Low-Starch Diet Group (LS): Participants in this group followed a low-starch diet, focusing on the consumption of low-glycemic-index vegetables, proteins, and healthy fats. Daily starch intake was limited to less than 50 grams [44].

Caloric Restriction Group (CR): Participants in this group followed a caloric restriction diet, reducing their daily caloric intake by 500 kcal from their estimated energy requirement, calculated based on the Harris–Benedict equation [45].

Control Group with Normal Weight (CTR): This group consisted of participants with normal weight who maintained their usual dietary habits without any specific diet intervention. Their caloric intake was not restricted or modified, serving as a baseline to assess natural variation in butyrate levels and other metabolic markers [46].

Dietary adherence was monitored through a weekly food diary and weekly online consultations with an endocrinologist.

### 4.3. Evaluation of Salivary Amylase Activity

Salivary amylase activity was measured using the Salimetrics Amylase Activity Assay (Salimetrics, State College, PA, USA). Unstimulated saliva samples were collected in the morning after a fast overnight to ensure consistency. Saliva samples were collected at baseline. The activity of saliva amylase was determined according to the manufacturer’s protocol. Participants were classified into high salivary amylase activity groups (HSA) and low salivary amylase activity (LSA) groups based on the median split of amylase activity data.

### 4.4. Measurement of Butyrate Levels

Blood samples were taken at the start of the study and at the end of the 12-week intervention period. Plasma butyrate concentrations were measured using the Butyric Acid ELISA Kit from Abbexa Ltd. Briefly, blood samples were centrifuged at 3000× *g* for 10 min at 4 ° C, and plasma was stored at −80 °C until analysis. Butyrate was extracted from plasma using an acidified water–ether solution (0.5 mL of 0.1 M hydrochloric acid in water mixed with 2.5 mL of ether). Butyrate levels were quantified based on calibration curves prepared using standard solutions [47].

### 4.5. Measurement of Insulin Sensitivity

Insulin sensitivity was assessed using the Homeostasis Model Assessment of Insulin Sensitivity (HOMA2-%S), which was calculated from fasting plasma glucose and C-peptide levels using the HOMA2 Calculator (Diabetes Trials Unit, University of Oxford).

### 4.6. Statistical Analysis

Data were analyzed using GraphPad Prism 10 software (GraphPad Software, San Diego, CA, USA). Given that the data did not follow a normal distribution, nonparametric tests were employed. The Shapiro–Wilk test was used to assess the normality of the data distributions. The Mann–Whitney U test was used to compare differences in basal butyrate levels between the high and low salivary amylase groups. Correlations between salivary amylase activity and butyrate concentrations were evaluated using Spearman’s rank correlation coefficient. A linear regression analysis was then performed to evaluate the predictive value of salivary amylase activity on the HOMA2-%S. The significance level was set at *p* < 0.05 for all analyses.

Descriptive statistics are presented as medians and interquartile ranges (IQRs) for continuous variables.

### 4.7. Ethics Statement

The study was carried out according to the Declaration of Helsinki and was approved by the Ethics Committee of Riga Stradiņš University (Ethical Committee number: 22-2/479/2021). Written informed consent was obtained from all participants prior to enrollment in the study.

## 5. Conclusions

This study demonstrates the link between salivary amylase activity, overweight, and glucose homeostasis. Baseline butyrate was higher in individuals with elevated salivary amylase activity, indicating a potential influence of this enzyme on gut microbiota metabolism. Post-intervention, butyrate production varied significantly across dietary groups, with low-starch diets and low salivary amylase activity showing increased butyrate levels, suggesting that salivary amylase activity influences gut microbiota composition, with differences observed depending on the type of diet. A weak but statistically significant correlation (ρ = 0.0486, *p* < 0.05) was found between salivary amylase activity and butyrate levels, supported by comparative tests showing differences in butyrate production across dietary patterns. Additionally, higher salivary amylase activity correlated with improved insulin sensitivity (β = 0.435, R^2^ = 0.35), highlighting its potential as a biomarker for metabolic health.

Future research should explore the mechanisms underlying these findings and validate them in larger, diverse populations to enhance personalized nutrition strategies targeting gut and metabolic health.

## Figures and Tables

**Figure 1 ijms-25-09956-f001:**
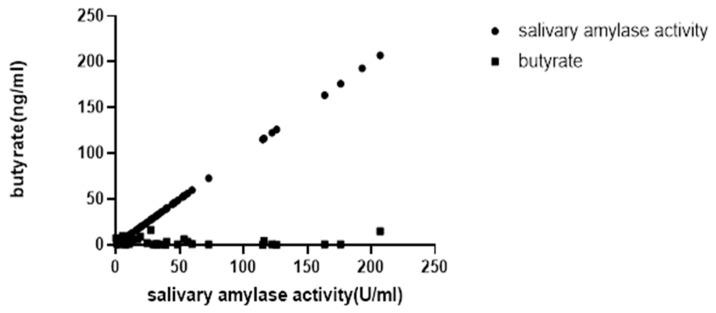
Correlation between salivary amylase activity and butyrate (*n* = 67). Spearman’s rank correlation coefficient indicated a significant but weak correlation between salivary amylase activity and butyrate (ρ = 0.0486, *p* < 0.05).

**Figure 2 ijms-25-09956-f002:**
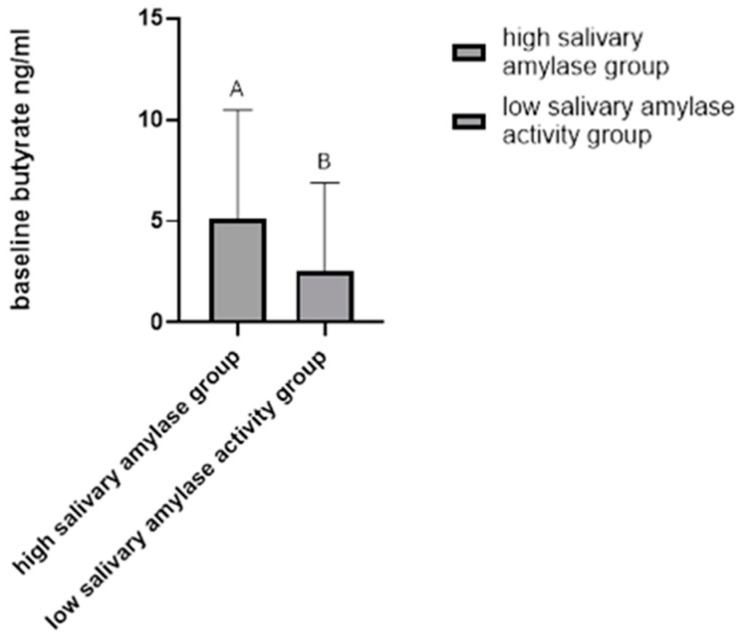
Basal butyrate level in participants with high vs. low salivary amylase activity. Women with high salivary amylase activity had significantly higher baseline butyrate levels compared to those with low salivary amylase activity (Mann–Whitney U = 44, *p* < 0.05). High salivary amylase (*n* = 30), low salivary amylase (*n* =30).

**Figure 3 ijms-25-09956-f003:**
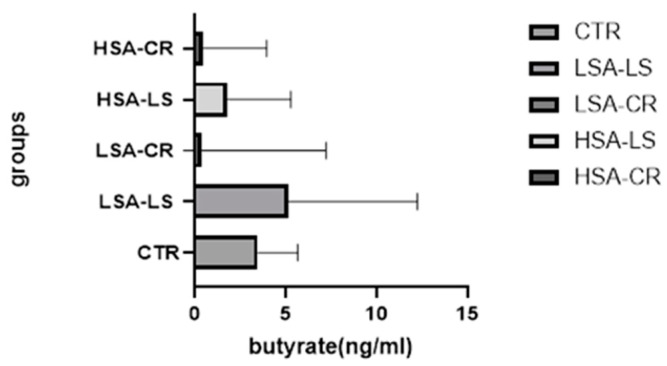
Butyrate levels after diet intervention by group. Low-starch diet with low salivary amylase activity (*n* = 15), low-starch diet with high salivary amylase activity (*n* = 15), calorie restriction diet with low salivary amylase activity (*n* = 15), calorie restriction diet with high salivary amylase activity (*n* = 15); control group (*n* = 7).

**Table 1 ijms-25-09956-t001:** Characteristics of the participants at baseline.

Group Allocation	Mean Age (Years) ± SD	Salivary Amylase (U/mL) Median (IQR)
Caloric Restriction (Low Salivary Amylase) (*n* = 15)	28.1 ± 3.9	15.3 ± 14.9
Low-Starch Diet (High Salivary Amylase) (*n* = 15)	29.4 ± 3.2	77.1 ± 50.5
Low-Starch Diet (Low Salivary Amylase) (*n* = 15)	28.5 ± 3.5	15.6 ± 19.7
Caloric Restriction (High Salivary Amylase) (*n* = 15)	30.1 ± 4.3	89.2 ± 47.4
Control Group (Normal Weight) (*n* = 7)	29.1 ± 3.2	31.7 ± 23.4

**Table 2 ijms-25-09956-t002:** Summary statistics of baseline key variables.

Variable	Value
Salivary Amylase (U/mL)	Median (IQR): 27.77 (10.64–56.24)
Butyrate (µmol/L)	Median (IQR): 1.823 (0.373–6.985)

**Table 3 ijms-25-09956-t003:** Key statistical results.

Group	Median Butyrate Level (ng/mL) and 95% CI for Median Values	IQR (ng/mL)
Low-Starch Diet (Low Salivary Amylase) (*n* = 15)	5.140	0.7070–12.25
Caloric Restriction (Low Salivary Amylase) (*n* = 15)	3.90	0.246–7.225
Caloric Restriction (High Salivary Amylase) (*n* = 15)	0.47	0.29–3.955
Low-Starch Diet (High Salivary Amylase) (*n* = 15)	1.8	1.13–5.29
Control Group (Normal Weight) (*n* = 7)	3.47 (95% CI: 0.14–7.15)	-
**Correlation**	**ρ**	** *p* ** **-value**
Salivary Amylase Activity and Butyrate	0.0486	*p* < 0.05
**Comparative Tests**	**Mann–Whitney U**	** *p* ** **-value**
High vs Low Salivary Amylase Activity	44	*p* < 0.05
Low-Starch Diet vs Caloric Restriction (Low Salivary Amylase)	59.50	*p* < 0.05

**Table 4 ijms-25-09956-t004:** The effect of age on the lower and higher IQRs of butyrate levels in each group.

Group	Mean Age (Years) ± SD	Lower IQR Butyrate Level (ng/mL)	Higher IQR Butyrate Level (ng/mL)
Low-Starch Diet (Low Salivary Amylase)	28.8 ± 2.57	0.7070	12.25
Caloric Restriction (Low Salivary Amylase)	28.43 ± 3.59	0.246	7.225
Caloric Restriction (High Salivary Amylase)	30.2 ± 4.47	0.29	3.955
Low-Starch Diet (High Salivary Amylase)	29.33 ± 3.67	1.13	5.29
Control Group (Normal Weight)	28.57 ± 2.57	0.951	5.67

**Table 5 ijms-25-09956-t005:** A linear regression analysis between salivary amylase activity and insulin sensitivity.

Predictor	β	SE	*p*-Value (β)	95% CI	R^2^	F (DF)	*p*-Value (Model)
Salivary Amylase Activity	0.435	0.12	<0.0003	[0.195, 0.675]	0.35	F (1, 34) = 5.5	0.0249

## Data Availability

Data are contained within the article.

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
