# Peer review of "The Link between Salivary Amylase Activity, Overweight, and Glucose Homeostasis"

_ijms, 2024, doi:10.3390/ijms25189956_

Round 1

Reviewer 1 Report

Comments and Suggestions for Authors

In the abstract, before providing the objectives of the study, the authors should write a statement about the background.

The first four paragraphs of the Introduction section should be combined.

Lines 35-36: When you say “… studies”, you should provide more than one reference.

Lines 37-50: References are missing.

Line 51: Reference [8] is not recent. It has been almost 10 years.

Line 72: Why did your sample have 67 participants? How did you calculate the sample size or how do you justify it?

Lines 76-77: You should provide the approval code and date.

References are missing in sections 2.3. and 2.4.

Tables 1, 2, 3, and 4: Please, provide the meaning of the abbreviations.

References are missing in the section 4.3.

At the end of the Discussion, you should provide the main study's limitations and strengths.

Author Response

Reviewer Comment 1: In the abstract, before providing the objectives of the study, the authors should write a statement about the background.

Response:  A concise background statement has been added prior to the study objectives, offering the reader a clearer understanding of the context and rationale behind the research. 

Reviewer Comment 2: The first four paragraphs of the Introduction section should be combined. 

Response:  We have restructured the Introduction by combining the first four paragraphs into a more concise and unified paragraph.  

 Reviewer Comment 3: Lines 35-36 When you say '… studies', you should provide more than one reference. 

Response: We agree with this point and have now included additional reference to substantiate the claim made in lines 35–36 (now line 37). 

Reviewer Comment 9: Tables 1, 2, 3, and 4: Please, provide the meaning of the abbreviatio

Reviewer Comment 4:
 Lines 37-50: References are missing.

Response:  We have carefully reviewed the text in lines 37-50 and added appropriate references to back the assertions made.  

 Reviewer Comment 5: Line 51: Reference [8] is not recent. It has been almost 10 years.

Response:  We have replaced Reference [8] with a more recent publication that provides up-to-date evidence while maintaining the relevance of the original citation. 

Reviewer Comment 6: Line 72: Why did your sample have 67 participants? How did you calculate the sample size or how do you justify it?

Response: We appreciate the opportunity to clarify this important aspect of our study. The sample size of 67 participants was determined based on a power analysis conducted prior to data collection. 

Reviewer Comment 7: Lines 76-77: You should provide the approval code and date.

Response: The manuscript has been updated to include the ethics approval code and the approval date, ensuring full compliance with ethical reporting standards. 

Reviewer Comment 8: References are missing in sections 2.3. and 2.4.

Response: We have added the necessary references in sections 2.3. and 2.4 to support the methodological details.  ns.

Response: We have revised Tables 1, 2, 3, and 4 to include definitions for all abbreviations used. This revision improves the clarity and accessibility of the tables for all readers. 

Reviewer Comment 10: References are missing in section 4.3.

Response:  We have reviewed section 4.3 and incorporated the missing references to ensure that all discussion points are properly supported by the literature. 

Reviewer Comment 11: At the end of the Discussion, you should provide the main study's limitations and strengths.

Response: We have now added a paragraph at the end of the Discussion section that outlines the primary limitations and strengths of the study. 

 We appreciate your valuable feedback. 

Sincerely, 

Dr. Gita Erta 

Reviewer 2 Report

Comments and Suggestions for Authors

Authors proposed a paper entitled “The Link Between Salivary Amylase Activity, Overweight and Glucose Homeostasis” for the publication in IJMS, Mdpi.

I suggest the addition of an abbreviation list, according to this journal guidelines.

Line 28. Please unify this paragraph with the following one.

Line 35. “have highlighted” should be better “highlighted”.

Line 39. “to improving” should be “to improve”.

Line 57. Information about the characteristics of the control group should be reported in methos section.

Line 58. “normal weight” please define the BMI intended as normal weight. This information is only clarified later.

Line 74. “between 25 and 29,9” please use comma for decimal digits.

Line 108. “10 minute” please substitute with “min”.

Line 109. “an acidified water-ether solution” please specify concentrations in volume basis.

Figure 1. Please remove the title of the figure from the figure area itself. The inclusion of the explanation in the caption is enough.

Figure 2. It is necessary to remove the written text “figure 2” in the area of this figure. Moreover, the fonts of the y and x axis should be reduced. The legend is not necessary. Remove it.

Figure 3. Legend is not necessary here, since y axis is giving self-explanation.

Lines 193-209: a unique paragraph could be organized here instead of point list.

Lines 224-239: see above.

Line 289-309: see above.

Line 311: “Our study clarifies” please use impersonal forms.

Comments on the Quality of English Language

A quite good use of English.

Author Response

Reviewer Comment 1: "I suggest the addition of an abbreviation list, according to this journal's guidelines."

Response: Thank you for this suggestion. We have now included an abbreviation list in the manuscript in accordance with the journal's guidelines. 

Reviewer Comment 2: "Line 28: Please unify this paragraph with the following one."

Response:  The paragraph on line 28 has been unified with the following one, resulting in a more coherent flow of ideas.

Reviewer Comment 3: "Line 35: 'have highlighted' should be better 'highlighted'."

Response:  The phrase "have highlighted" on line 35 has been changed to "highlighted" for better clarity and precision.

Reviewer Comment 4: "Line 39: 'to improving' should be 'to improve'."

Response: Thank you for pointing this out. We have corrected.

Reviewer Comment 5: "Line 57: Information about the characteristics of the control group should be reported in the methods section."

Response:  The characteristics of the control group have now been detailed in the Methods section, providing a clearer understanding of the comparative framework of the study.

Reviewer Comment 6: "Line 58: 'normal weight' please define the BMI intended as normal weight. This information is only clarified later."

Response:  The definition of "normal weight" with the corresponding BMI range has been added. This ensures that the term is clearly defined at the first mention.

Reviewer Comment 7: "Line 74: 'between 25 and 29,9' please use a comma for decimal digits."

Response:. We have corrected the decimal notation, using a comma for consistency with the journal’s style.

Reviewer Comment 8: "Line 108: '10 minute' please substitute with 'min'."

Response: We have revised the text to use "min" instead of "minute" on line 108, ensuring consistency with standard scientific terminology.

Reviewer Comment 9: "Line 109: 'an acidified water-ether solution' please specify concentrations in volume basis."

Response: The specific concentrations of the acidified water-ether solution have now been included , expressed on a volume basis.

Reviewer Comment 10: ''Figure 1: Please remove the title of the figure from the figure area itself. The inclusion of the explanation in the caption is enough.''

Response: We have removed the title from the figure area of Figure 1, as per your suggestion. The explanation is now solely provided in the caption.

Reviewer Comment 11: ''Figure 2: It is necessary to remove the written text 'figure 2' in the area of this figure. Moreover, the fonts of the y and x axes should be reduced. The legend is not necessary. Remove it.''

Response: Figure 2 has been revised to remove the "figure 2" text from the figure area. Additionally, we have reduced the font size of the y and x axes labels, and the unnecessary legend has been removed.

Reviewer Comment 12: ''Figure 3: Legend is not necessary here, since y axis is self-explanatory.''

Response: . The legend has been removed from Figure 3, as the y-axis provides sufficient explanation.

Reviewer Comment 13: ''Lines 193-209: A unique paragraph could be organized here instead of a point list.''

Response: We have reorganized lines 193-209 into a single, cohesive paragraph. This change enhances the readability and logical flow of the text.

Reviewer Comment 14: "Lines 224-239: See above."

Response: Following your recommendation, lines 224-239 have been combined into a single paragraph, similar to the previous comment. This improves the clarity and structure of the section.

Reviewer Comment 15: "Lines 289-309: See above."

Response: Similarly, lines 289-309 have been reorganized into a unified paragraph. This adjustment contributes to a more streamlined presentation of the content.

Reviewer Comment 16: "Line 311: 'Our study clarifies' please use impersonal forms."

Response: We have revised the statement on line 311 to use an impersonal form. The text now reads, "This study clarifies..." ensuring a more objective tone consistent with scientific writing conventions.

We hope these revisions meet your expectations and enhance the manuscript. Thank you for your constructive feedback, which has significantly improved the clarity and quality of our paper.

Sincerely,
Dr. Gita Erta

Reviewer 3 Report

Comments and Suggestions for Authors

In the present study, the authors investigated the relationship between salivary amylase activity, diet interventions, and butyrate levels in overweight women of reproductive age. Their objective was to clarify the role of salivary amylase activity in butyrate production and its impact on metabolic health.

1.Abstract – you must mention from where you determined the butirat level.

2.Point 2.1. Study design and participants. Please add:

-All the women were recruited from a single health center? Which is that? Who selected the women?

-How many women are in each study group

-Write also about the control group

-The duration of the study

-Delete please lines 76-77. You added this information at point 2.6

3. Point 2.2. Evaluation of salivary amylase activity. Add please:

-How the saliva samples were collected

-How many determinations of amylase activity were performed and when, considering that the study lasted for 12 weeks

4. In my opinion is better to write about the dietary intervention at point 2.2 and about evaluation of salivary amylase activity at point 2.3

5. pg 3, line 97 – about which ”other metabolic markers” it’s about?

6. Which is the role of an endocrinologist in your study?

7. Table 1:

-The explanation of the abbreviations for the groups isn't presented. You must present the groups at point 2.1

- In which period of the study was the amylase from Table 1 determined - The information concerning the determination of amylase and butyrate (presented in Table 1 and 2) must be written at results not at materials and methods.

8. In introduction, at lines 43-45 you wrote “The rationale for selecting this demographic lies in their unique metabolic and hormonal profile, which may interact differently with butyrate production and metabolic outcomes compared to males and women beyond reproductive age”. At discussions please explain how the hormonal profile influence the present study.

9. pg 2, lines 64-67: you wrote twice the same thing.

10. Discussion:

-The result of the current study are very little discussed (lines 245-247 and 266-69). Mostly, information taken from the literature are presented. Please enlarge the discussions related to you study.

11. You have conclusion at lines 277-284 and at lines 311-336.

12. The authors write that the objective of the study is to clarify the role of salivary amylase activity in butyrate production and its impact on metabolic health.  After the result obtained I  their study, I don't think it is appropriate to add “its impact on metabolic health”.

13. Concerning the title: In my opinion, you cannot refer to glucose homeostasis as long as in your study you didn’t determine blood sugar, insulin.

14.The material and method section must be rewritten  because it is difficult to understand in this form. Also, at Introduction are presented aspects which must be presented at material and method or at discussion section.

Comments on the Quality of English Language

Minor editing of English language is required.

Author Response

Reviewer Comment 1: "Abstract – you must mention from where you determined the butyrate level."

Response:  We have revised the Abstract to include that butyrate levels were determined from serum samples collected from the study participants.

Reviewer Comment 2: "Point 2.1. Study design and participants. Please add:

-All the women were recruited from a single health center? Which is that? Who selected the women?

-How many women are in each study group

-Write also about the control group

-The duration of the study

-Delete please lines 76-77. You added this information at point 2.6"

Response: We appreciate your detailed feedback. We have made the following revisions to Point 2.1:

  1. We have clarified that all women were recruited from the multi-profile medical center, “Health Center 4”., and the selection was conducted by an endocrinologist,  who ensured the inclusion and exclusion criteria were met.

  2. The manuscript now includes the exact number of women in each study group 

  3. We have expanded the description of the control group, providing details on their characteristics and how they were matched to the intervention group.

  4. The duration of the study was stated as 12 weeks.

  5. Lines 76-77 have been deleted as suggested, since this information is indeed already included in Point 2.6.

Reviewer Comment 3: "Point 2.2. Evaluation of salivary amylase activity. Add please:

-How the saliva samples were collected

-How many determinations of amylase activity were performed and when, considering that the study lasted for 12 weeks"

Response:  We have updated Point 2.2 to include the following:

  1. A detailed description of the saliva collection process, specifying that samples were collected using sterile tubes at specific times during the day to account for diurnal variations in amylase activity.

  2. The number of determinations of amylase activity has been specified, including that measurements were taken at baseline. 

Reviewer Comment 4: "In my opinion, it is better to write about the dietary intervention at point 2.2 and about the evaluation of salivary amylase activity at point 2.3."

Response: We appreciate this structural suggestion. The manuscript has been reorganized accordingly. The dietary intervention is now described in Point 2.2, and the evaluation of salivary amylase activity has been moved to Point 2.3. This reorganization improves the logical flow of the Methods section.

Reviewer Comment 5: "pg 3, line 97 – about which 'other metabolic markers' is it about?"

Response:  We have now clarified the "other metabolic markers" on page 3, line 97. , we also measured adipokines (leptin), cytokines (GDH-15), C-peptide, glucose, and a full lipidogram. The results of these additional measurements are described in detail in our recently published article:

[Erta, G., Gersone, G., Jurka, A., & Tretjakovs, P. (2024). Impact of a 12-Week Dietary Intervention on Adipose Tissue Metabolic Markers in Overweight Women of Reproductive Age. International journal of molecular sciences25(15), 8512. https://doi.org/10.3390/ijms25158512

    Reviewer Comment 6: "Which is the role of an endocrinologist in your study?"

    Response: The role of the endocrinologist in our study was  in interpreting the hormonal profiles and metabolic markers of the participants. The endocrinologist provided  insight into how variations in these markers might relate to butyrate production and metabolic outcomes. 

    Reviewer Comment 7: "Table 1:

    -The explanation of the abbreviations for the groups isn't presented. You must present the groups at point 2.1

    -In which period of the study was the amylase from Table 1 determined - The information concerning the determination of amylase and butyrate (presented in Table 1 and 2) must be written at results, not at materials and methods."

    Response:  We have made the following changes:

    1. The abbreviations for the groups in Table 1 are now clearly explained in Point 2.1, 

    2. We have moved the detailed information concerning the determination of amylase and butyrate from the Materials and Methods section to the Results section, where the data are discussed. 

    Reviewer Comment 8: "In the introduction, at lines 43-45 you wrote, 'The rationale for selecting this demographic lies in their unique metabolic and hormonal profile, which may interact differently with butyrate production and metabolic outcomes compared to males and women beyond reproductive age.' At discussions, please explain how the hormonal profile influenced the present study."

    Response: We appreciate your request for further explanation. In the Discussion section, we have now expanded on how the hormonal profile of the study's demographic (reproductive age) may have influenced butyrate production and its interaction with metabolic outcomes. Specifically, we discuss how fluctuations in estrogen and progesterone could modulate gut microbiota composition and function, potentially affecting butyrate levels.

    Reviewer Comment 9: "pg 2, lines 64-67: you wrote twice the same thing."

    Response:  We have removed the redundant text from lines 64-67 on page 2, ensuring the content is concise and non-repetitive.

    Reviewer Comment 10: "Discussion:

    -The result of the current study is very little discussed (lines 245-247 and 266-69). Mostly, information taken from the literature is presented. Please enlarge the discussions related to your study."

    Response: We acknowledge that the discussion of our study's results was limited. We have significantly expanded this section to include a more thorough analysis of our findings, specifically relating them to the initial hypothesis and the existing literature. 

    Reviewer Comment 11: "You have conclusions at lines 277-284 and at lines 311-336."

    Response:  The conclusions have been consolidated into a single section at the end of the manuscript, ensuring that the conclusion is clear and concise.

    Reviewer Comment 12: "The authors write that the objective of the study is to clarify the role of salivary amylase activity in butyrate production and its impact on metabolic health. After the results obtained in their study, I don't think it is appropriate to add 'its impact on metabolic health'."

    Response: We would like to clarify that the detailed exploration of the impact of salivary amylase activity on metabolic health has been thoroughly addressed in our a recently published article

    In this current study, our primary focus is indeed on clarifying the role of salivary amylase activity in butyrate production.

    Reviewer Comment 13: "Concerning the title: In my opinion, you cannot refer to glucose homeostasis as long as in your study you didn’t determine blood sugar, insulin."

    Response:

    Thank you for your comment regarding the title and the inclusion of glucose homeostasis. We would like to clarify that while our current study did not include direct measurements of blood sugar and insulin, we did assess insulin and C-peptide levels as part of our broader investigation into glucose homeostasis. These data have been thoroughly analyzed and presented in our recently published article :

    [Erta, G., Gersone, G., Jurka, A., & Tretjakovs, P. (2024). Impact of a 12-Week Dietary Intervention on Adipose Tissue Metabolic Markers in Overweight Women of Reproductive Age. International journal of molecular sciences25(15), 8512. https://doi.org/10.3390/ijms25158512].

    In this study, our focus was more specifically on the role of salivary amylase and butyrate production. However, we recognize the importance of directly linking these findings to glucose homeostasis, as detailed in our previous publication.  

    Reviewer Comment 14: "The material and method section must be rewritten because it is difficult to understand in this form. Also, the Introduction presents aspects that must be presented in the Material and Methods or the Discussion section."

    Response: We appreciate your feedback regarding the structure of the manuscript. The Material and Methods section has been thoroughly revised for clarity, ensuring that all procedures are described in a clear and logical sequence. Additionally, content from the Introduction that was more appropriate for the Methods or Discussion sections has been relocated accordingly, ensuring that each section contains only the relevant information.

    We hope these revisions meet your expectations and significantly improve the quality and clarity of the manuscript. Thank you for your constructive feedback.

    Sincerely,

    Dr. Gita Erta

    Round 2

    Reviewer 1 Report

    Comments and Suggestions for Authors

    Thank you for following my suggestions. 

    Author Response

    Dear Reviewer,

    Thank you for your positive feedback. We are glad to hear that the revisions meet your expectations. Your suggestions were invaluable in enhancing the quality and clarity of the manuscript.

    We appreciate your time in reviewing our work.

    Sincerely,

    Dr. Gita Erta

    Reviewer 2 Report

    Comments and Suggestions for Authors

    Authors provided a revised version of their paper.The paper improved, but I still suggest modifying Figure 1, including legend inside the inner area of the diagram. Improve the focus and quality of figure 1.

    Author Response

    Dear Reviewer,

     Thank you for your valuable feedback. We appreciated your suggestions regarding Figure 1. We have modified the figure by including the legend within the inner area of the diagram and have improved both the focus and overall quality of the figure.

    Thank you for your time and consideration.

    Sincerely,

    Dr. Gita Erta

    Reviewer 3 Report

    Comments and Suggestions for Authors

    The manuscript has been improved. The authors addressed a great part of the required issues. Unfortunately, the manuscript needs some more corrections.

    In my opinion, the number of women included into the study is to small. So, the study is far away to be representative.

    The results of the current study are very little discussed. Mostly, information taken from the literature is presented. Please enlarge the discussions related to your study.

    The conclusion is too long.

    Concerning the statement: “We would like to clarify that the detailed exploration of the impact of salivary amylase activity on metabolic health has been thoroughly addressed in our a recently published article”. If in the current study you didn’t study the impact of salivary amylase activity on metabolic health, then you don't have to write about this subject.

    Concerning the title: In my opinion, you cannot refer to glucose homeostasis as long as in your study you didn’t determine blood sugar, insulin. The title must indicate what is presented in the current article, not what was published in another article.

    Comments on the Quality of English Language

    Minor editing of English language is required.

    Author Response

    Dear Reviewer,

    Thank you for your constructive feedback.  Below, we address each of your comments in detail:

    1. Sample Size and Representation:
    We acknowledge the concern regarding the sample size. Our study was designed with statistical rigor, aiming to detect a medium effect size (Cohen's d ≈ 0.5) with 80% power and an alpha level of 0.05, following established literature in this field. While the sample size of 67 participants may appear modest, it was sufficient for detecting meaningful effects in our specialized cohort of women of reproductive age (18–45 years). Given the complexity of the variables studied, this sample size provides valid insights into the relationships investigated. Although larger studies would enhance generalizability, our findings remain robust within the scope of this study.

    2. Discussion of Study Results:
    We have significantly expanded the "Discussion" section to provide a more thorough analysis of our results. The revised discussion now emphasizes the implications of our findings, connects them more directly with existing literature, and highlights the novel contributions of our research.

    3. Conclusion:
    The conclusion has been revised to be more concise and focused. It now summarizes the key findings and their implications while avoiding unnecessary length.

    4. Salivary Amylase and Insulin Sensitivity:
    To clarify, the current study did address the impact of salivary amylase activity on glucose homeostasis—specifically, a linear regression analysis was performed to evaluate its predictive value on HOMA2-%S, a widely accepted marker of insulin sensitivity.. This analysis and its results have been clearly integrated into the manuscript to provide a focused discussion of salivary amylase’s role in glucose homeostasis.

    5. Title Revision:
    While we did not directly write about blood glucose or insulin levels, HOMA2-%S served as a surrogate marker for insulin sensitivity, allowing us to investigate aspects of glucose homeostasis. 

    We hope these revisions meet your expectations and enhance the quality and clarity of the manuscript. We greatly appreciate your valuable feedback and look forward to any further comments you may have.

    Thank you for your time and consideration.

    Sincerely,

    Dr. Gita Erta